# Usability and feasibility of a digital cognitive screening tool measuring older adults' early postoperative neurocognitive recovery: a protocol for a pilot study

Anahita Amirpour [1], Jeanette Eckerblad,[1] Anders Thorell,[2] Lina Bergman [1], Ulrica Nilsson [1,3]

¹Neurobiology, Care Sciences and Society, Karolinska Institutet, Stockholm, Sweden
²Department of Clinical Sciences Intervention and Technology, Karolinska institutet, Huddinge, Sweden
³Perioperative Medicine and Intensive Care, Karolinska Universitetssjukhuset, Stockholm, Sweden

**Correspondence to**
Anahita Amirpour;
anahita.amirpour@ki.se

## ABSTRACT

**Introduction** Delayed neurocognitive recovery, also identified as early postoperative cognitive decline (POCD), is a common complication after surgery, with advanced age being the most important risk factor. As the geriatric population is increasing worldwide, and number of older adults undergoing surgery continues to rise, so will the incidence of POCD. Only a small proportion use digital cognitive tests for measuring postoperative neurocognitive performance compared with analogue tests. This study aims to evaluate a digital cognitive screening tool, Mindmore Postoperative version (Mindmore-P), in a perioperative setting to determine its feasibility and usability, and to compare preoperative cognition with early postoperative neurocognitive performance. Further, to determine associations between neurocognitive performance and perioperative factors as well as to explore patients' experiences of early neurocognitive recovery.

**Methods and analysis** We will include 50 patients (aged ≥60 years) undergoing elective abdominal surgery under general anaesthesia. Cognitive functions will be measured with Mindmore-P preoperatively and on postoperative day (POD) 1 or 2 as well as 2–3 weeks after surgery. Preoperatively, frailty, (Clinical Frailty Scale), depression (Geriatric Depression Scale-15), functional status (12-item WHO Disability Assessment Schedule 2.0) and pre-recovery status (Swedish web version Quality of Recovery Scale, SwQoR) will be measured. Delirium will be assessed by Nu-DESC (Nursing Delirium Screening Scale) twice a day, with start on POD 1 and until the patient is discharged from the hospital. Outcomes at 2–3 weeks postoperatively are postoperative recovery (SwQoR), depression, functional status and usability (System Usability Scale) of Mindmore-P. Postoperative recovery will also be measured POD 1 or 2. We will also explore feasibility and experience of early postoperative neurocognitive recovery with interviews approximately 1 month after surgery.

**Ethics and dissemination** This study is approved by the Swedish Ethical Review Authority (REC Reference: 2022-03593-01) and will follow the principles outlined in the 1964 Helsinki Declaration and its later amendments. Results from this study will be disseminated in peer-reviewed journals, scientific conferences and in social media.

**Trial registration number** NCT05564195.

### STRENGTHS AND LIMITATIONS OF THIS STUDY

⇒ This study follows the Medical Research Council's framework for designing and evaluating complex interventions.
⇒ This study will provide evidence for the feasibility and usability of a digital neurocognitive tool to be used in a population of older persons undergoing surgery.
⇒ This pilot study will create a knowledge base for future interventions as previous studies are lacking.
⇒ A limitation is that the study only includes two follow-ups.

## INTRODUCTION

The geriatric population is increasing worldwide.[1] Every fifth person in Sweden today is ≥65 years, and the number of persons of 80 years and over is estimated to increase by 50% until 2050.[2 3] National statistics show that approximately 700 000 people undergo inpatient surgery, and almost 50% of those are performed on people ≥60 years.[4] However, hospital care after surgery remains short for the afflicted population; hence a major part of the postoperative recovery occurs after discharge from the hospital which[2] poses several risks for older people such as cognitive decline.[1]

Cognitive decline in older individuals has become an area of great concern, particularly given the impact of cognitive impairment on functional capacity.[5] Cognitive functions in late adulthood decline at different pace, and some cognitive domains are more prone to faster decline, such as executive functions,

speed processing and memory.[6] Delayed neurocognitive recovery, earlier identified as postoperative cognitive decline (POCD), is an early postoperative cognitive complication, included in perioperative neurocognitive disorders. Delayed neurocognitive recovery is diagnosed up to 30 days after anaesthesia and surgery[1 7] and is characterised by a gradual decreased performance in several cognitive domains such as memory, speed processing, executive function and attention.[1 8] The term POCD will be used in this article when referring to earlier research.

Preoperative risk factors for developing POCD after surgery are advanced age, pre-existing cognitive decline, visual and audial impairment, ongoing infection or serious illness.[9] Other risk factors are impaired functional status, lower level of educational attainment,[7 10 11] alcohol abuse, electrolyte disturbances that are either medication-induced or due to dehydration, psychoactive medications, depression and frailty.[10 11] Although no causality has been demonstrated, POCD is associated with poor postoperative outcomes including prolonged length of hospital stay, increased morbidity/mortality and reduced quality of life, resulting in a significant burden on the individual, their family, as well as for the healthcare system. To date there is no existing pharmacological or non-pharmacological intervention to treat postoperative neurocognitive decline or decrease the associated symptoms.[7 11] Moreover, results have been inconclusive regarding the role of intraoperative factors such as type of anaesthesia, hypotension, cerebral saturation or choice of anaesthetic drugs for the development of POCD.[12] One hypothetical mechanism is that older adults have an unregulated acute inflammatory response to surgery.[13] So far, an explicit connection between POCD and neuroinflammation due to the surgical trauma has been confirmed.[7 14] The American Geriatrics Society's guidelines[9] recommend that every geriatric perioperative patient with risk factors, should be assessed for their cognitive performance prior surgery. Despite the potential benefits, it seems challenging to implement such a routine in regular clinical practice. There is therefore a need to develop and implement feasible measures to assess cognitive function in a busy clinical environment.[11]

The methods used for measuring postoperative neurocognitive performance vary widely between studies, with only a small proportion, 13.5%, using digital cognitive tests or a combination of analogue and digital tests.[15] To cover this gap we conducted a comparability study[16] between the digital cognitive test battery Mindmore Postoperative version (Mindmore-P), and the analogue International Study Group of Postoperative Cognitive Dysfunction (ISPOCD) test battery.[17] The study included 50 cognitively healthy participants aged 60 years and older. We found that the included tests in ISPOCD and Mindmore-P were comparable. Further, the participants were overall positive to digital testing and thought that neurocognitive testing was useful and important to assess if they would undergo surgery (Manuscript in progress). The next step, following Medical Research Council's framework for designing and evaluating complex interventions,[18] is to assess predefined progression criteria to reduce uncertainty around recruitment, data collection, outcomes, sample size and analysis in a future intervention study.

## Aim
The aim of this study is to pilot test Mindmore-P in a clinical perioperative setting. To achieve this, we intend:
1. To test the recruitment process and measure attrition rate.
2. To determine the usability and feasibility of Mindmore-P.
3. To explore the trajectory of delayed neurocognitive recovery by measuring the patient's preoperative and postoperative neurocognitive performance.
4. To investigate if there are any associations between the dependent variable; delayed neurocognitive recovery and the independent variables; delirium, preoperative frailty, symptoms of depression, postoperative delirium, physical function, and early postoperative recovery.
5. To determine patients' experiences of early neurocognitive recovery.

## METHODS AND ANALYSIS
### Study design
This is an observational feasibility study.

### Participants, recruitment, and setting
We will include 50 patients undergoing elective abdominal surgery under general anaesthesia at Ersta Hospital in Stockholm, Sweden. Patient recruitment is planned to start in December 2022 and end in July 2024. The sample size is guided by a reported incidence of early cognitive decline of approximately 13%–25% at 1–2 weeks postoperatively.[17] At the patient's preoperative outpatient visit, a research nurse will provide oral and written information about the study. The details of the study and its potential benefits and risks will be explained carefully to the patient. If the patients agree to participate in the study, they will undergo a Mini-Mental State Examination (MMSE) screening with the research nurse to examine if they are eligible for the study.

### Inclusion criteria
1. ≥60 years of age undergoing elective non-cardiac surgery with general anaesthesia.
2. Duration of surgery >60 min
3. MMSE score >24.

### Exclusion criteria
1. MMSE score <23 indicating cognitive impairment.[19]
2. Inability to read and speak Swedish.
3. Uncorrected severe visual or auditory disorder, disease of the central nervous system, psychiatric diseases including alcohol abuse or drug dependence, current

**Table 1** Included tests and cognitive domains in the Mindmore-P

| Cognitive domain | Mindmore-P |
|---|---|
| Verbal episodic memory | Consortium to Establish a Registry for Alzheimer's Disease (CERAD) A 10-word verbal learning test with three learning trials, a recall trial after ~7 min. Words are read and visually presented on the screen and the patient replies in speech. |
| Executive function, visuospatial function | Trail Making Test (TMT-A and B) Each part consists of 25 circles with letters or numbers on the screen. In part A, the circles are numbered from 1 to 25. The patient is instructed to draw a line between them in ascending order of numbers, from 1 to 25, as rapidly as possible. Part B consists of 13 digits and 12 letters to be paired, and the test subject is instructed to draw a line between them in ascending number and letter order. |
| Executive function, selective attention | Stroop Colour-Word Test (SCW) 24 words spelling out the name of a colour which is printed in contrasting ink colours (eg, the word 'green' printed with red ink), and the participant is asked to tell the printed colour of the word rather than the actual meaning of the word. |
| Executive function, visuospatial function | Symbol Digits Processing Test (SDPT) The test consists of nine symbol–digit pairings at the top of the screen, one symbol in the middle of the screen and a 3×3 number grid at the bottom of the screen. The participant is required to associate the symbol in the middle to one of the digits on the grid below. As soon as a choice is made a new symbol appears. The final score is the number of correct matches in 90 s |

Mindmore-P, Mindmore Postoperative version.

motor impairment in dominant hand, and colour vision deficiency.

## Outcomes
### Cognitive tests
The intervention is the digital cognitive test battery Mindmore-P (table 1), which will assess neurocognitive performance. The test battery measures the following cognitive domains: verbal episodic memory, executive functions, visuospatial function and attention.

The Mindmore-P is self-administered on a touch screen tablet (10.1″ 20–30 min Windows), and is a web application running on a full-screen Chrome browser. The test is administered by using a capacitive touchscreen with features that record detailed information such as timing, pauses and lifting of finger from the touchscreen. Speech recognition is used in the screening of verbal memory and language. Mindmore's platform is CE certified and includes normative data adjusted for age, education and sex[20]

### Usability
The usability (table 2) of Mindmore-P will be assessed with the System Usability Scale (SUS). SUS consists of 10 items to be answered on a 5-point scale from 'strongly agree' to 'strongly disagree'.[21]

### Depression screening
The Geriatric Depression Scale (GDS) is a widely used instrument to screen for depression in the older adult population.[22] The GDS-15 consists of 15 items with binary answer (yes/no) to questions such as '*Do you feel helpless?*' or '*Are you in good spirits most of the time?*'.[23] The cut-off score is standardised with a score of 6 to the Swedish older adult population.[24] Every participant will answer the Swedish version of the GDS-15 before surgery and on postoperative day (POD) 14 during the revisit at the hospital.

### Preoperative frailty
Preoperative frailty will be measured by using the 9-item Clinical Frailty Scale (CFS), translated to a Swedish version, the scale ranges from 1 *very vital* to 9 *terminally ill*. The Swedish version has been validated in persons >65 years seeking care at an emergency department.[25]

### Delirium assessment
Delirium assessment will be performed using the observational 5-item scale Nu-DESC, where symptoms are rated from 0 to 2. A score of 2 or more indicates delirium.[26] A Swedish version of the Nu-DESC has been validated in persons undergoing cardiac surgery.[27]

### Postoperative recovery
Postoperative recovery will be measured with Swedish web version of Quality of Recovery (SwQoR). SwQoR is a psychometric evaluated including 24 different symptoms such as fatigue, nausea, anxiety, headache, pain in the surgical wound, fever, to be rated on 11-point numerical scales from 0 'never' to 10 'always'.[28] Global score ranging from 0 'excellent quality of postoperative recovery/no symptoms' to 240 'extremely poor quality of recovery/ severe symptoms', with cut-off values less than 31 at POD 7 and less than 21 on POD 14 indicating good recovery.[29] Reliability and validity tests have presented evidence that it is suitable to use SwQoR in patients undergoing surgery.[28]

### Functional capacity
Functional status (table 2) will be measured with the Swedish version of 12-item WHO Disability Assessment Schedule 2.0 (WHODAS), ranges from 0 to 48, with increasing scores reflecting increasing disability.[30]

### Demographic and perioperative data
These include age, sex, educational level, MMSE score, comorbidities, American Society of Anaesthesiologists

**Table 2** Outcome measures

| Outcome measures | Questionnaires | Preop | POD 1 or POD 2 | Revisit at hospital POD 14+0–5 days | One month postop |
|---|---|---|---|---|---|
| Dependent variable | | | | | |
| Postoperative neurocognitive recovery | Mindmore-P | X | X | X | |
| Independent variables | | | | | |
| Frailty | Clinical Frailty Scale | X | | | |
| Depression | GDS-15 | X | | X | |
| Functional capacity | WHODAS-12 | X | | X | |
| Daily delirium assessment with start on day of surgery | Nu-DESC | | | | |
| Postoperative recovery | SwQoR | X | X | X | |
| Usability of Mindmore-P | SUS | | | X | |
| Feasibility: experiences of neurocognitive recovery and digital assessment | Qualitative interviews | | | | X |

GDS-15, Geriatric Depression Scale-15; Mindmore-P, Mindmore Postoperative version; POD, postoperative day; SUS, System Usability Scale; SwQoR, Swedish web version of Quality of Recovery; WHODAS-12, 12-item WHO Disability Assessment Schedule 2.0.

classification, premedication, choice of anaesthetic drugs, duration of anaesthesia and surgery, type of anaesthesia and surgery, intraoperative bleeding (mL), postoperative pain scores (numeric rating scale), intraoperative and postoperative opioids, early postoperative complications, and length of hospital stay.

## Feasibility and patients' experiences of early postoperative cognitive recovery

Approximately 20–25 of the study participants will be included in an individual face-to-face interview exploring patients' experiences of early neurocognitive recovery and the use of Mindmore-P. The sample size will follow the concept of information power and will be determined within the process of data collection. Information power means that the more information the sample has, the lower the need to increase the amount of participants.[31] A semistructured interview guide will be constructed and tested; examples on questions from the interview guide are '*Can you tell me about how you have been since you were discharged from the hospital?*' To gain a deeper understanding, probing questions will be asked, such as: '*Have you experienced any changes in your memory since the surgery?*', '*how has it been to plan and do different tasks after the surgery?*' and '*you mentioned …*', '*Could you tell me more about that?*', '*how do you mean* and *how would you describe that?*' and '*Can you give me any example?*'. The participants will be asked to speak freely about their experiences of their postoperative neurocognitive recovery. All interviews will be audiotaped.

## Procedure

### Preoperative data collection

Eligible patients who consent to participate in the study will undergo the cognitive Mindmore-P test battery preoperatively. Before the test battery starts, we will inform about the Mindmore-P procedure and during the first test session a trained research nurse will be present in the room to assist with any technical help needed. Each test consists of an introduction both in audio and visualised on-screen, and the instructions can be repeated on demand. A practice trial is performed before the test battery starts. The digital cognitive test is expected to take less than 30 min. Depression, frailty, postoperative recovery (except for three questions about the surgical wound) and functional status will be measured after the cognitive testing is completed by the trained research nurses (table 2). It is estimated to take 15–20 min to complete the questionnaires. The day and time of day for preoperative data collection will be documented.

### Postoperative data collection

We will assess delirium (Nu-DESC) twice a day (morning and afternoon) with start on POD 1 and until the patient is discharged from the hospital. Postoperative neurocognitive performance (Mindmore-P) and postoperative recovery (SwQoR) will be measured on POD 1 or 2, and at the follow-up visit approximately 2 weeks after the surgery. We will measure depression (GDS-15), functional status (WHODAS) and usability of Mindmore-P (SUS) at the follow-up visit. Feasibility and experience of postoperative neurocognitive recovery will be assessed with interviews approximately 1 month after surgery in the patient's home or at the hospital (table 2).

### Analysis

Descriptive statistics of demographic data will be presented with number, percentage and mean (SD) or range, as appropriate. A within-subject analysis and a standardised reliable change Z-score for each postoperative neurocognitive test (n=4) will be calculated, based on the difference between patient's test results at baseline (i.e., preoperative) versus postoperative tests results, and normalised using Mindmore's normative data for

each specific test.[20] As recommended by Borchers *et al* review from 2021, mild POCD (1–2 SD) and major POCD (>2 SD) will be specified for each test result and POCD is present if at least two tests indicate decline.[15]

Usability and attrition rates will be presented with descriptive statistics. Demographic characteristics, potential cofounders (i.e., patients characteristics, surgical and anaesthesiologic factors), preoperative depression (GDS-15), frailty (CFS), postoperative recovery (SwQoR) and functional status (WHODAS) will be compared between the groups, that is, delayed neurocognitive recovery or not, using parametric and non-parametric tests. To analyse intraindividual differences over time in cognitive performance, the same statistical tests as well as multivariate logistic regression analysis will be used. A p-value of <0.05 will indicate statistical significance in all analyses. We will analyse the interviews with thematic analysis.[32]

### Ethics and dissemination

All participants will obtain information about consent, confidentiality, voluntary basis of the study and the participant can refuse participation at any time. Each study participant will receive detailed information about duration and extent of the digital cognitive tests and the included outcomes, by the trained research nurses. The study participant can refuse to participate based on this information. Written consent will be obtained by the trained research nurses after they have received written and verbal information about the study. All study-related information will be stored securely digitally and physically at Ersta Hospital and at the university site. The local databases will be secured with password-protected access systems. The trial investigators will have access to the data; at present there is no plan to grant public access to participant-level dataset and statistical code after the study is completed. The project has been approved by the Swedish Ethical Review Authority (REC Reference: 2022-03593-01) and will follow the principles outlined in the 1964 Helsinki Declaration and its later amendments. Results from this study will be disseminated in peer-reviewed journals, scientific conferences and in social media. The trial investigators will follow the Uniform Requirements for Manuscripts.[33]

### Patient and public involvement

Patients or the public were not involved in the design, or conduct, or reporting, or dissemination plans of this research.

### DISCUSSION

An evidence-based, feasible and usable digital neurocognitive tool is needed, especially as the growing population of older adults is increasing, and thereby the incidence of POCD is likely to rise. Both the American Geriatrics society[9] and The American Society of Anesthesiologists[1] recommend that older adults should be assessed for their cognitive performance prior to and after surgery.

In line with this recommendation, we have conducted a comparability study between the analogue ISPOCD test battery and the Mindmore-P digital test battery. The different test batteries were comparable, and the participants were overall positive to the digital cognitive test and thought the tests were useful and important to take if they would undergo surgery (Amirpour, Eckerblad, Bergman, Nilsson, 2023). We will now take the next step evaluating Mindmore-P in a new context, that is, the clinical perioperative setting. This is an evaluation and adaptation process that follows the Medical Research Council's framework for designing and evaluating complex interventions.[18] Following a systematic process when measuring a complication that is not currently measured in clinical practice, that is, postoperative neurocognitive performance, increases the transferability and quality of the evidence that will be produced. This pilot study is therefore of great importance to conduct as it will determine not only the feasibility and usefulness of Mindmore-P, but it will also guide us in the choice of independent variables, the primary outcome and the sample size in a future study.

**Contributors** AA, JE, AT, LB and UN contributed to planning of the study, study design and the preparation of the manuscript and approved the final version.

**Funding** This work was supported by grants from the Strategic Research Area Health Care Science (grant number 2021-01095), The Erling-Persson Foundation (grant number 20190200) and Karolinska Institutet (grant number 2020-02641). These funding sources had no role in the design of the study and will have no role in its execution, analysis, data interpretation, or decision to submit results.

**Competing interests** None declared.

**Patient and public involvement** Patients and/or the public were not involved in the design, or conduct, or reporting or dissemination plans of this research.

**Patient consent for publication** Consent obtained directly from patient(s).

**Provenance and peer review** Not commissioned; externally peer reviewed.

**ORCID iDs**
Anahita Amirpour http://orcid.org/0000-0002-2939-2592
Lina Bergman http://orcid.org/0000-0002-7510-8679
Ulrica Nilsson http://orcid.org/0000-0001-5403-4183

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
