## [Reviewer comments · BMJ Open]

ARTICLE DETAILS

TITLE (PROVISIONAL)	Usability and feasibility of a digital cognitive screening tool measuring older adults' early postoperative neurocognitive recovery: A protocol for a pilot study
AUTHORS	Amirpour, Anahita; Eckerblad, Jeanette; Thorell, A; Bergman, Lina; Nilsson, Ulrica

VERSION 1 – REVIEW

REVIEWER	Jiang, Yuan Chengdu Medical College
REVIEW RETURNED	09-Dec-2022

GENERAL COMMENTS	In this protocol, the authors tried to evaluate the feasibility and usability of Mindmore-P in a perioperative setting. They want to compare preoperative cognition with early postoperative neurocognitive performance. Further, they want to determine associations between neurocognitive performance and perioperative factors as well as explore patients' experiences of early neurocognitive recovery. There are some concerns. If the authors can't add some explanation to the revised manuscript, I think it would not be accepted. Introduction part 1. "...a major part of the postoperative recovery occurs after discharge from the hospital which poses several risks for older people... What risk for older people ?2. The authors should introduce the Mindmore-P. What is the perfect score for Mindmore-P? What is the criterion score for diagnosing POCD?3. "... We found that the included tests in ISPOCD and Mindmore-P were comparable..." In the previously published protocol of the authors about the comparison between ISPOCD and Mindmore-P, 50 cognitively healthy older would be employed (doi: 10.1136/bmjopen-2022-062007). We do not know how to use ISPOCD and Mindmore-P to evaluate the elderly with cognitive health and get comparable results between ISPOCD and Mindmore-P in this protocol. Except for the difference between handwriting and tablet, we can not see the big difference between ISPOCD and Mindmore-P in content detection. We are even worried about the reliability and validity of Mindmore-P in evaluating POCD. Methods and Analysis part 4. The inclusion criteria are very rough. POCD often occurs in elderly patients undergoing major cardiac surgery. Why did the author not choose elderly patients undergoing cardiac surgery?
--

	5. The score of cognitive impairment defined by MMSE is related to the educational level of patients, so why there is no requirement for the educational level of patients in this study? 6. It has been reported that the probability of POCD in the elderly over 75 years old is much higher than that in the elderly over 65 years old. Will the different ages of the subjects affect the evaluation results of Mindmore-P? 7. If the patient has postoperative delirium or gives up on the subject, does he need to be excluded? 8. Is the computer interface of Mindmore-P friendly to the elderly? Analysis part 9. "...mild POCD (1-2 standard deviation [sd]) and major POCD (>2 sd) will be specified for each test result and POCD is present if at least two tests indicate decline..." Whether the test is Mindmore-P. We do not know what the score standard of Mindmore-P is, the Z value? 10. Did the authors consider using serological indicators or EEG to evaluate POCD?
--	--

REVIEWER	Guttenthaler, Vera Univertätsklinik Bonn, Anaesthesiology
REVIEW RETURNED	13-Dec-2022

GENERAL COMMENTS	This is a very interesting feasibility study that addresses a major topic. To get the most reliable data maybe the timepoint of follow-up should be specified more clearly, e.g. 14 days +/- 2 days so that the follow ups are comparable. If it is feasible it would be interesting to ask patients to answer the Mindmore-P questionnaire at the 30-day follow up one again to get a broader view of the patients´ postoperative neurocognitive recovery. The timepoint of postoperative recovery assessment should be at the same time for all patients to establish best comparability. The question is whether the QoR questionnaire is suitable to compare patients health status before and after surgery. Maybe it would be better to choose an instrument that could be used in both settings e.g. EQ5D-5L to compare the quality of life before and after surgery and use the QoR only after surgery to assess the recovery. Usually the results of the QoR questionnaire are interpreted in that way that a higher score means a better quality of recovery. This might confuse some readers and should be clearly explained.
--

VERSION 1 – AUTHOR RESPONSE

Reviewer: 1

Dr. Yuan Jiang, Chengdu Medical College

Comments to the Author:

In this protocol, the authors tried to evaluate the feasibility and usability of Mindmore-P in a perioperative setting. They want to compare preoperative cognition with early postoperative neurocognitive performance. Further, they want to determine associations between neurocognitive performance and perioperative factors as well as explore patients´ experiences of early neurocognitive recovery.

There are some concerns. If the authors can't add some explanation to the revised manuscript, I think

it would not be accepted.

Introduction part

1. "...a major part of the postoperative recovery occurs after discharge from the hospital which poses several risks for older people... What risk for older people ?

- We refer to risk of cognitive decline. This information is now given in the revised manuscript, Page 5, line 91.

2. The authors should introduce the Mindmore-P. What is the perfect score for Mindmore-P? What is the criterion score for diagnosing POCD?

-The a perfect score for Mindmore -P has been clarified on page 12, line 277-81: A within-subject analysis a standardized reliable change Z-score for each postoperative neurocognitive test (n=4) will be calculated, based on the difference between patient's test results at baseline at baseline (i.e. preoperative) vs. postoperative tests results, and normalize using Mindmore's normative data for each specific test.

- Information about the criterion score for diagnosing POCD (mild POCD will be set at 1-2 standard deviation [sd] and major POCD at >2 sd will be specified for each test result and POCD is present if at least two tests indicates decline) can be find on page 12, line 281-83

3. "... We found that the included tests in ISPOCD and Mindmore-P were comparable..." In the previously published protocol of the authors about the comparison between ISPOCD and Mindmore-P, 50 cognitively healthy older would be employed (doi: 10.1136/bmjopen-2022-062007). We do not know how to use ISPOCD and Mindmore-P to evaluate the elderly with cognitive health and get comparable results between ISPOCD and Mindmore-P in this protocol. Except for the difference between handwriting and tablet, we can not see the big difference between ISPOCD and Mindmore-P in content detection. We are even worried about the reliability and validity of Mindmore-P in evaluating POCD.

-The reviewer refers to a published protocol of ours. We understand that it is not possible to find the requested information in that protocol since it do not include any results. The results we are referring to, in the submitted manuscript, is a manuscript in progress i.e. unpublished results. Page 6, line 128-32.

Methods and Analysis part

4. The inclusion criteria are very rough. POCD often occurs in elderly patients undergoing major cardiac surgery. Why did the author not choose elderly patients undergoing cardiac surgery?

- POCD has been more frequently evaluated in patients undergoing cardiac surgery and we therefore know that this is a big problem in this group. However, what we know is that perioperative triggering factors for POCD includes long duration of surgery such as in cardiac surgery but it also appears after other procedures such as abdominal surgery.

5. The score of cognitive impairment defined by MMSE is related to the educational level of patients, so why there is no requirement for the educational level of patients in this study?

- We collect data on education level. See Demographic and perioperative data Page 10.

6. It has been reported that the probability of POCD in the elderly over 75 years old is much higher than that in the elderly over 65 years old. Will the different ages of the subjects affect the evaluation results of Mindmore-P?

- Information about normative data and the adjustments of the test results has been inserted on page 8, line 188-89 and page 12, line 277-81.

7. If the patient has postoperative delirium or gives up on the subject, does he need to be excluded?

- No, they will not be excluded.

8. Is the computer interface of Mindmore-P friendly to the elderly?

- Yes. In our manuscript in progress, the older adult participants found it highly user-friendly. Moreover, Mindmore-P normative data include persons of < 20 to > 80 years.

Analysis part

9. "...mild POCD (1-2 standard deviation [sd]) and major POCD (>2 sd) will be specified for each test result and POCD is present if at least two tests indicate decline..." Whether the test is Mindmore-P.

We do not know what the score standard of Mindmore-P is, the Z value?

- The information that Mindmore's normative data been inserted on page 8, line 188-89 and page 12, line 277-81.

10. . Did the authors consider using serological indicators or EEG to evaluate POCD?

- Inflammatory biomarkers has been included in another study of ours. We will not include any intraoperative EEG measurement such as BIS in this study as the evidence of its effect on POCD is weak.

Reviewer: 2

Dr. Vera Guttenthaler, Univeritätsklinik Bonn Comments to the Author:

This is a very interesting feasibility study that addresses a major topic. To get the most reliable data maybe the timepoint of follow-up should be specified more clearly, e.g. 14 days +/- 2 days so that the follow ups are comparable.

- The information has been updated to postoperative day 14 + 0-5 days, Table 2. Early postoperative neurocognitive decline (POCD) appears from 7 to 30 days postoperatively. It may be a weakness that we do not assess neurocognitive function on exactly the same day for all participants. However, there is no evidence that cognitive function differs within a week but does differ over a longer period of time postoperatively and, in particular, between patients' pre- and post-operative function.

If it is feasible it would be interesting to ask patients to answer the Mindmore-P questionnaire at the 30-day follow up one again to get a broader view of the patients' postoperative neurocognitive recovery. The timepoint of postoperative recovery assessment should be at the same time for all patients to establish best comparability.

- Yes, it would be interesting to measure neurocognitive function at 30 days as well, but we decided to assess twice postoperatively as this is "only" an observational feasibility study. A study that will create a base for a future study guiding us in the the recruitment process, attrition rate, the usability and feasibility of Mindmore-P but as well as guide us in the choice of independent variables, the primary outcome and the sample size in a future study. A clarification has been inserted on page 613, line 332-34

The question is whether the QoR questionnaire is suitable to compare patients health status before and after surgery. Maybe it would be better to choose an instrument that could be used in both settings e.g. EQ5D-5L to compare the quality of life before and after surgery and use the QoR only after surgery to assess the recovery.

- The QoR can indeed be use pre- as well as postoperatively as most of the questions (21/24) are symptoms that also can be present before surgery, such as anxiety, sleeping difficulties, fatigue etc. It's only the questions regarding the surgical wound (n=3) that has to be omitted. This information has been inserted in the manuscript. Page 9, line 213-17 and page 10, line 256-57

Usually, the results of the QoR questionnaire are interpreted in that way that a higher score means a better quality of recovery. This might confuse some readers and should be clearly explained.

- We agree with the reviewer, and this has now been clarified in the methods section. Page 9, line 214-18.

VERSION 2 – REVIEW

REVIEWER	Guttenthaler, Vera Univertätsklinik Bonn, Anaesthesiology
REVIEW RETURNED	11-Apr-2023
GENERAL COMMENTS	There are some small spelling (anaesthesia - anesthesia) inconsistencies and some sentences have to be revised (line 43/44, line 277-281 and line 318).